# The Role of Lactoferrin in Intestinal Health

**DOI:** 10.3390/pharmaceutics15061569

**Published:** 2023-05-23

**Authors:** Celia Conesa, Andrea Bellés, Laura Grasa, Lourdes Sánchez

**Affiliations:** 1Departamento de Producción Animal y Ciencia de los Alimentos, Facultad de Veterinaria, Universidad de Zaragoza, 50013 Zaragoza, Spain; cconesam@gmail.com; 2Departamento de Farmacología, Fisiología y Medicina Legal y Forense, Facultad de Veterinaria, Universidad de Zaragoza, 50013 Zaragoza, Spain; a.belles@unizar.es (A.B.); lgralo@unizar.es (L.G.); 3Instituto Agroalimentario de Aragón IA2 (UNIZAR-CITA), 50013 Zaragoza, Spain; 4Instituto de Investigación Sanitaria de Aragón (IIS Aragón), 50009 Zaragoza, Spain

**Keywords:** lactoferrin, intestinal health, iron absorption, intestinal growth, intestinal maturation, intestinal damage repair, immune system, antimicrobial activity, microbiota

## Abstract

The intestine represents one of the first barriers where microorganisms and environmental antigens come into tight contact with the host immune system. A healthy intestine is essential for the well-being of humans and animals. The period after birth is a very important phase of development, as the infant moves from a protected environment in the uterus to one with many of unknown antigens and pathogens. In that period, mother’s milk plays an important role, as it contains an abundance of biologically active components. Among these components, the iron-binding glycoprotein, lactoferrin (LF), has demonstrated a variety of important benefits in infants and adults, including the promotion of intestinal health. This review article aims to provide a compilation of all the information related to LF and intestinal health, in infants and adults.

## 1. Introduction

Gastrointestinal (GI) health is an important pillar of the overall health and well-being of animals and humans. The GI tract plays a role that extends beyond the digestion and absorption of nutrients. Healthy GI development and function also impacts the immune system. The intestine is a barrier that prevents the entry of infectious pathogens, protects the host against systemic immune responses to commensal bacteria and food antigens, and prevents the trillions of commensal microorganisms living in the gut from reaching systemic sites [1].

The early postnatal phase is of vital importance to development, as the infant moves from the uterus, where it is protected, to an environment with unfamiliar antigens and pathogens. The GI tract goes through a major adaptation, in both structure and functionality, to allow the digestion and absorption of nutrients from the diet. Human milk contains an abundance of biologically active components, such as bioactive proteins, lipids, and carbohydrates, which are more than likely to play a key role in the intestinal benefits of breastfeeding. For the past 30 years, preclinical and clinical studies have further investigated the iron-binding glycoprotein, lactoferrin (LF), because of its wide range of biological activities, both in infants and adults [2]. Those activities include the modulation of immunity and inflammation; antioxidant, anti-tumour, and antimicrobial activities; enhancement of iron absorption; and the elicitation of cellular responses, including activation, differentiation, and proliferation [3].

The aim of this review has been to compile the effects that LF can exert, at the intestinal level, in infants and adults, including iron intestinal absorption; intestinal growth, maturation, and damage repair; regulation of the innate and adaptive immune system; antimicrobial activity; as well as affecting the microbiota (Figure 1).

## 2. Lactoferrin and Intestinal Iron Absorption

It is well known that iron is a vital nutrient at any period of life. Studies have shown that levels above and below the optimum recommended for this metal can have serious and lifelong consequences. In this manner, insufficient iron can cause irreversible damage of central nervous system during development of young mammals, and an excess of iron can have adverse long-term implications, probably due to excessive production of free radicals [4]. Neither adults nor mammalian animals are known to have a mechanism to regulate excretion of iron and then, the level of iron in the body is determined by its absorption in the small intestine [5].

Iron, naturally present in foods and beverages, is inefficiently absorbed in the intestine, and this absorption varies depending on dietary factors and other factors related to the host. It is estimated that only about 10–20% of the iron taken orally is absorbed from the diet [6].

As reviewed by Artym et al. [7], and summarized here (Figure 2), iron ions are obtained from the diet by enterocytes in both the duodenum and jejunum, after which they are transported by the carrier protein—divalent metal transporter 1 (DMT-1). Only Fe^2+^ ions can enter into enterocytes being necessary to reduced Fe^3+^ ions present in the diet by the action of duodenal cytochrome b (Dcytb). Heme Fe^2+^ ions from meat are transported to enterocytes via the heme carrier protein 1 (HCP-1) and are released into the cytosol through heme oxygenase 1 (HO-1). The Fe^2+^ ions in enterocytes can either be temporarily stored in the iron-binding protein—ferritin (Ftn), or be released into the circulation by the basolateral membrane transporter—ferroportin (FPN), which is the only exporter of iron ions in cells. Subsequently, there are two oxidases, membrane hephaestin and serum ceruloplasmin (Cp), which oxidize Fe^2+^ to Fe^3+^ ions. Then, Fe^3+^ ions are transported in the blood by transferrin (TF) to every cell and tissue in the body. The expression of both iron transporters mentioned previously (DMT-1 and FPN) is regulated by the available systemic iron stores. Therefore, when there is a high level of iron, their expression decreases, and increases again when reserves are reduced. In addition, iron homeostasis is regulated at the systemic level by the hepatic hormone, hepcidin that is a central regulator of systemic iron. When hepcidin binds to FPN, it causes the internalization and degradation of this iron transporter, and the release of iron from the intestine into circulation is inhibited.

There is a lot of evidence demonstrating that LF is a regulator of bodily iron metabolism. LF in the gut (coming from the diet or endogenously released) may act as an iron supplier to intestinal cells, although the mechanism of this action is still not clear [7]. Several studies have demonstrated the binding of LF to Caco-2 cells, a cell line that, when differentiated, acts as a human intestinal epithelium model. The European Medicines Agency (EMA) and the Food and Drug Administration (FDA) recognize the use of this cell line as an in vitro intestinal model, to estimate the absorption, metabolism, and bioavailability of different drugs or xenobiotics following oral ingestion [8,9]. The binding of LF to differentiated Caco-2 cells has been demonstrated in several studies for LF from human milk, recombinant human LF (rhLF) from Aspergillus awamori, rhLF from rice [10,11], and bLF [12]. Another proposal was that LF would bind to a membrane protein or region, releasing the iron on the cell surface, and subsequently delivering it to the cell via a non-vesicular transport pathway, as it occurs in other cells. Several studies have shown that the binding of LF to intestinal cells is due to electrostatic interactions, as this binding can be blocked by other cationic proteins such as lactoperoxidase [10,11]. On the other hand, the binding of human LF (hLF) to an LF receptor (LFR) present in the human small intestine apical membrane has been demonstrated. The receptor has been further studied by characterization, cloning, and sequencing. It has been found to be identical to intelectin, which is a carbohydrate-binding protein of the intestine [13]. Intelectin is known to bind LF and other molecules such as the carbohydrate chains of the bacterial cell wall that contains galactofuranose [14,15]. However, it does not bind to other glycosylated proteins, such as transferrin, nor to other non-glycosylated proteins such as the bovine serum albumin [16].

The research of Mikogami et al. [17] suggests that the synthesis of LFRs in intestinal epithelial cells is regulated in response to intracellular iron levels. Therefore, in case of intracellular iron depletion, LFR synthesis is induced and thus, it increases the LF iron uptake in enterocytes. Hence, LFR may play a role in iron absorption, mainly in the case of iron deficiency [17], since there is higher iron absorption when there is a higher number of receptors. This interrelationship is of particular importance during iron absorption in newborns and infants, because the mechanisms of classic regulation by the membrane transporters, DMT-1 and FPN, are poorly developed. Accordingly, the LF contained in breast milk or formula milk must guarantee an iron supply adequate to the current needs of the intestine, protecting against iron deficiency or excess [7,18].

Independently of the way in which iron is taken from LF, it has been shown that hLF can be internalized by Caco-2 cells and that both the protein and Fe are taken up (Figure 2) [10,11,13]. In vitro studies with bLF have shown that it is not fully degraded in the luminal environment of the stomach and at the level of the small intestine, allowing it to bind to brush border membrane receptors [19]. These receptors mediate the bLF uptake into intestinal cells in a specific manner. bLF is transported from the intestinal lumen to the bloodstream, and acts both at the intestinal luminal level and at the systemic level [6].

In a study carried out in LF knockout (LFKO) mice, it was demonstrated that, when compared with the wild-type (WT), LF was not essential for iron supply during the postnatal period. Other studies on adult mice, fed with a basal or a high-iron diet, showed no differences in either transferrin saturation or tissue iron reserves between the WT and LFKO mice, although serum iron levels were moderately elevated in LFKO mice with the basal diet. In situ hybridization analysis showed that LF is not expressed in the postnatal or adult intestine, which is consistent with the relatively normal iron status. These results provide evidence that LF apparently does not play an important role in the regulation of iron homeostasis in animals [20].

On the other hand, studies in humans have also been performed with different results. In a study by Davidsson et al. [21], no direct role of LF in iron absorption was found, as breastfed children had lower levels of Fe than the children fed with infant formula without LF. However, in the research performed by Rosa and Trugo [22], it was concluded that LF can increase iron absorption during the neonatal period, which contributes to an elevated bioavailability of this mineral in human milk. A study with human volunteers showed that the nutritional importance of LF may not be limited to infants, as it is also a bioavailable source of iron, equivalent to ferrous sulphate, in young adult women [23]. As it has been widely reviewed by others [7,24], LF supplementation has been found to be effective in the treatment of iron deficiency disorders. The oral administration of LF normalizes iron homeostasis in humans, not only promoting iron absorption, but also inhibiting the inflammatory processes causing anemia in chronic pathologies.

Anemia has already become a global public health concern [25]. Pregnant women and children are the main susceptible population, as they need a high iron intake for fetal and bodily development [26]. As reviewed by Paesano et al. [27], oral administration of iron, in the form of ferrous sulphate, is the most widely used treatment for dealing with iron deficiency (ID) and ID anemia (IDA). However, the use of ferrous sulphate as a treatment frequently fails to exert the desired effect on these pathologies, and sometimes causes several adverse effects including gastrointestinal distress, nausea, vomiting, diarrhea or constipation. The poor bio-availability of inorganic iron, which requires the administration of a large amount of ferrous sulphate in order to take effect, may be the cause of these disorders. Due to the ability of LF to reduce the expression of pro-inflammatory cytokine in vivo, including IL-6 (reviewed below), it could potentially represent a promising novel natural compound for use in the prevention and cure of pregnancy-associated ID and IDA. In this way, oral administration of bLF could be an effective therapy to prevent and cure ID and IDA during pregnancy [28]. In this work, which represents the first clinical trial on the therapeutic potential of bLF on systemic iron homeostasis, the efficacy of this natural compound, in restoring iron homeostasis in pregnant women suffering from ID and IDA, was demonstrated. Regardless of the trimester of pregnancy, oral administration of 100 mg of LF (about 30% iron-saturated), twice daily before meals, increased total serum iron and hemoglobin concentrations to a larger extent than that found after administration of oral ferrous sulphate. Unlike the administration of ferrous sulphate, oral administration of LF did not produce any side effects [28]. The strong effect shown by LF, compared with ferrous sulphate, was neither due to the amount of iron provided by LF (8.8 mg/day), lower than that of ferrous sulphate (156 mg/day) [28], nor to the enhanced iron absorption, as the absorption of iron from LF and ferrous sulphate was found to be equal [23].

In another clinical trial, which involved 143 pregnant women with ID and IDA, oral administration of LF was also found to increase red blood cells counts and serum ferritin concentrations to a greater extent than orally administered ferrous sulphate [27]. In an attempt to obtain more information on the therapeutic potential of orally administered LF in the control of systemic iron homeostasis, this research group performed a study on five pregnant women with ID and IDA. These women were treated for 30 days with LF, followed by another 30 days of treatment with ferrous sulphate. The numbers of red blood cells were analyzed, as well as the concentrations of hemoglobin, total iron, ferritin, and IL-6 in serum. All blood parameters analyzed increased after treatment with LF, while these values decreased after 30 days of treatment with ferrous sulphate. Regarding IL-6, its concentration decreased significantly after 30 days of treatment with LF, while it increased again after 30 days of treatment with ferrous sulphate [27]. Oral administration of ferrous sulphate in healthy volunteers, receiving 120 mg of iron per day, also showed an inflammatory response [29]. Elevated IL-6 levels correlated well with low values of total serum iron and serum ferritin. In contrast, the low IL-6 levels, found after LF treatment, were well correlated with the increase in total serum iron and serum ferritin values [27].

Similar results were found in a clinical trial, conducted on 29 children by Mikulic et al. [30], in which the amount of iron absorbed from holo-LF (iron-saturated form) was comparable to that of ferrous sulphate. Moreover, adding apo-LF (iron-depleted form) to a test meal containing ferrous sulphate, significantly increased the absorption of iron (+56%). These results suggested that LF facilitates iron absorption in young infants. Since LF binds iron with high affinity, it could be a safe approach to provide iron to infants in low-income countries, where iron fortification can adversely affect the intestinal microbiome and lead to diarrhea. In another randomized placebo-controlled intervention trial in 260 infants, carried out by Chen et al. [31], there was a significantly lower incidence rate of respiratory and diarrheal illnesses, and fewer symptoms of rhinorrhea, coughing, and wheezing after 3 months of being fed with bLF-enhanced formula, thus indicating that LF may be efficient not only in enhancing iron absorption, but also in reducing diarrhea in iron-supplemented infants.

A recent systematic review and meta-analysis of clinical studies was performed by Zhao et al. [24], who investigated the way LF modulates iron metabolism and assessed the comparative effects between LF and ferrous sulphate supplementation on iron absorption, iron storage, erythropoiesis, and inflammation. Regarding their findings, LF is considered a better supplement than ferrous sulphate, due to the enhancement of serum iron parameters and hemoglobin levels. Since LF appears to have little influence on iron absorption, the anti-inflammatory effect of LF might be the mechanism which explains its effectiveness on iron status and erythropoiesis. In this study, it is concluded that LF supplementation has better effects on serum iron, ferritin, and hemoglobin concentration. Moreover, LF reduces iron absorption and IL-6 levels, compared with ferrous sulphate.

Bovine milk-derived LF (bLF) was approved as novel food ingredient by the European Food Safety Authority (EFSA) in 2012 [32], and earlier, as Generally Recognized as Safe (GRAS) for use in infant and adult as a food additive and dietary supplement by the Food and Drug Administration (FDA) in 2001 [33]. Numerous in vitro animal and human studies have confirmed the safety and tolerability of bLF. Results of acute, sub-chronic and chronic oral toxicological assays in rats showed that LF is well tolerated with no adverse effects. Moreover, neither carcinogenicity nor genotoxicity has been detected. A substantial body of scientific evidence from published intervention studies provides support for the safety of bLF in humans, including infants and children [7,34]. For the above reasons, and for its easy production, bLF is the type of LF selected to be used as Fe supplement.

## 3. Lactoferrin and Intestinal Growth, Maturation and Damage Repair

Breastfeeding is known to be associated with health benefits, as, for example, lower incidence of neonatal necrotizing enterocolitis (NEC) and diarrhea during the childhood period [35,36]. The WHO recommends exclusive breastfeeding for 6 months and complementary breastfeeding for up to 2 years of age and beyond. However, due to the nutritional and health status of some mothers, resulting from various social factors or due to other circumstances, many newborns cannot be breastfed, so infant formula is undoubtedly the best substitute in these situations. Nowadays, based on the premise that infant formula should be as similar as possible to breast milk, it is based on milk protein, and contains mainly whey protein, rather than casein, as in the past. Supplementation of infant formula with LF, which is found in high levels in breast milk, deserves more attention [35,37]. In fact, large-scale manufacturing of bLF, from bovine skim milk and whey, was established more than 30 years ago. The pioneers in the production of bLF, on an industrial scale, were the Oleofina Company in Belgium and MILEI GmbH in Germany (using technology developed by Morinaga Milk Industry Co., Ltd.) [38]. bLF has been added as a supplement to several products in Japan since the 1990s, including infant formula, yogurt, specialized milk-based and other beverages, nutritional supplements, pet foods, and cosmetics [39]. Infant formulas enriched with bLF are also available in other countries, including Indonesia, South Korea and, as mentioned previously, bLF was approved as Generally Recognized as Safe (GRAS) for use in infants and adults, as a food additive and dietary supplement by the Food and Drug Administration (FDA) in 2001 [33], as well as as a novel food ingredient by the European Food Safety Authority (EFSA) in 2012 [32].

### 3.1. Lactoferrin and Intestinal Growth and Maturation

The significant growth and development of intestinal mucosa during the first years of life is the result of an equilibrium between proliferation and differentiation. The balance between proliferation and programmed cell death (apoptosis) produces a controlled increase in cell population. This balance is especially important in the small intestine, where continuous processes of cell elimination and replacement occur [40]. LF has been suggested to be a gut development modulator, through direct effects on intestinal epithelial cell proliferation and differentiation during infancy.

The postnatal development of intestinal epithelium has a long-term impact on the healthy growth of infants, and the nutritional composition of the diet plays an important role in it. hLF facilitates the intestinal adaptation of infants. Hydrolysis of LF is minimal in infants, due to gastric pH [41], and, therefore, LF might have greater biological potential in infants than in adults [40].

In vivo and in vitro studies have aided the study of the positive effects of LF on the intestine. Regarding in vitro studies, native hLF (with around 10% iron saturation) was observed to bidirectionally regulate cellular growth in Caco-2 cells in a dose-dependent manner [42]. When high LF concentrations were used, there was a strong increase in intestinal epithelial cell proliferation, whereas when using low LF concentrations, there was an activation of epithelial cell differentiation, as was indicated by the upregulation of sucrase and lactase activities (Figure 3). This bidirectional activity of LF confers flexibility, according to the need of the environment and its changing concentration in breast milk during lactation (higher concentrations at the beginning of lactation) [43]. A study by Buccigrossi et al. [42] mimicked the two types of lacteal secretions with different LF concentrations, colostrum and mature milk. It was observed that the higher concentration of LF in colostrum helps the accumulation of small intestinal tissue mass, whereas the main function of LF, at lower concentrations in mature milk, was mainly directed to the proper differentiation of progenitor cells to one of its functional cell types in the small intestine. Moreover, in the work of Blais et al. [44], LF was found to increase Caco-2 cell proliferation in a dose-dependent manner. Higher cell density and an increase in cell differentiation indicators were obtained, such as brush-border-associated enzymatic activities or amino acid transporters, and were increased with different kinetics. LF was also noted to be able to decrease Caco-2 cell spontaneous apoptosis. bLF was identified as being able to promote cell growth and arrest cell-cycle progression in the G2/M-phase and also, on human intestinal epithelial crypt cells (HIECs), which are more similar to the cells in the human small intestine [19].

It has been shown, as mentioned in the previous section, that LFR of the Caco-2 cell membrane is responsible for the uptake of apo- and holo-LF, through clathrin-mediated endocytosis, and this is related to the process of proliferation and differentiation [45]. The complex of LF and LFR is internalized through clathrin-mediated endocytosis. Both apo-LF and holo-LF activate the PI3K/Akt pathway, whereas only apo-LF triggers ERK1/2 signaling. LF, when entering the nucleus, can stimulate the incorporation of thymidine into crypt cells, regulating the transcription of genes as TGFβ1 [40]. Using HIEC cell model, hLF and bLF isolated from raw milk, and commercial bLF, were found to promote proliferation of these cells and also their development via a serial of mechanisms, such as by initiating the signaling of RhoA, Wnt/b-catenin, ERK/MAPK, telomerase, and growth hormone [46].

Several studies have shown that LF, at different degrees of iron saturation, can induce different biological effects, depending on the cell type. In that way, Oguchi et al. [47] reported that holo-LF induced proliferation on confluent (i.e, mature) Caco-2 cells, whereas apo-LF suppressed it. In contrast, in subconfluent non-differentiated Caco-2 cells, only apo-LF, but not holo-LF, stimulated the proliferation of Caco-2 cells [42,45]. Since proliferative mechanisms are likely to be diminished in differentiated Caco-2 cells, the effect of LF on differentiated cells could be reasonably different from the effect on undifferentiated highly proliferative cells. Iron saturation of LF in human milk is around 10% and remains the same during lactation [43]. The fact that in human milk, the major part of LF is not iron-saturated, may have physiological relevance and can be related to the role of LF in the intestinal development of infants [45].

Moreover, LF has been shown to stimulate the activity of both lactase and sucrase in subconfluent Caco-2 cells, when added at an early stage of differentiation [42]. In view of these results, it can be deduced that LF has a positive role in the regulation of lactase activity levels in human milk, since its increase, during the first weeks of life in preterm infants, was greater than the increase in the growth of small-intestinal mucosa [48].

There are several in vivo studies performed to determine the role of LF on the growth and development of intestinal cells. The effect of hLF in the milk of transgenic mice was assayed on the intestinal growth in suckling neonates. The results indicated that intestinal growth was increased in neonates that received milk containing hLF. Small intestine weight increased by about 27% and the intestinal length increased by about 6.5%, suggesting that milk LF increased mucosal growth. Moreover, the maltase to lactase ratio (marker of maturation) in the duodenal segment of the small intestine was also significantly higher in the offspring that suckled transgenic milk. These findings confirm that oral intake of hLF enhances the growth and maturation of intestinal mucosa, and indicates a potential therapeutic role of LF in premature infants and in patients with intestinal diseases, such as Crohn’s disease [49].

Some studies performed in mice have shown that supplementation with bLF enhanced jejunal villus height and the expression of various enzymes of the intestinal brush border membrane [44]. These stimulatory effects on cell growth and differentiation were related to a decrease in the apoptosis of cells. Increased intestinal cell growth and differentiation was related to the presence of LF in the gastrointestinal tract and in peripheral blood when mice were fed an LF-supplemented diet. In that mouse model, LF would either act from the basolateral site or from the apical site of the intestinal epithelium to promote epithelial cell growth. Those findings revealed that an increase in mouse intestinal cell proliferation and differentiation was correlated with a decrease in TGF-β receptor and caspase-3 expression, which is consistent with earlier studies, showing that TGF-β signaling pathways are implicated in the regulation of intestinal cell division, differentiation, adhesion, migration, and death [40]. Reducing apoptosis of the gut-differentiated cells would thus increase their survival rate [44].

As mentioned previously, Wang et al. [37] found that milk with exogenous LF supplied to mice, in contrast to LF-free milk, from LF gene systemic knockout mice, exerted a positive effect on intestinal development and maturation. They noted that the intestinal density, maturity, and barrier integrity of mice drinking milk without LF were inferior to those of mice drinking milk with LF. In that way, after analyzing the three segments of the small intestine in mice drinking milk without LF, maltase/lactase activity, as an indicator of intestinal maturity, showed that its maturity was lower than that of mice fed milk with LF. In the control group, the maturity of the small intestine was reduced from the duodenum to the ileum, which would indicate an increase in LF digestion in this section of the intestine. The jejunum is the principal portion of the small intestine, in terms of digestion and absorption, and lactation is a critical period for the development of this part of the intestine, in order to satisfy the newborn’s nutritional needs. Those results indicated that LF intake during the lactation period had the most significant effect on the jejunum. LF exerted a significant role on the jejunal villus/crypt in suckling mice. LF also enhanced, in the jejunum, the expression of an indicator of barrier integrity, the occludin, but had no significant effects in other sections of the small intestine.

In a study carried out on weaned piglets, LF supplementation resulted in an increase of 15.3% in villus height in the intestine, compared to the control group (no LF supplementation) [50]. Moreover, another in vivo study, conducted by Reznikov et al. [51], showed that bLF stimulated crypt cell proliferation in neonatal pigs. In piglets fed a bLF-containing formula for the first 14 days of life, jejunal intestinal crypt proliferation, and crypt depth and area were inceased. In addition, the cells of the jejunal crypt, isolated by laser capture microdissection (LCM), increased the expression of β-catenin mRNA. The increased expression of β-catenin suggests that the Wnt-signaling transduction pathway may partly mediate the stimulatory effect of bLF on intestinal cell proliferation. Moreover, there was no significant effect of dietary bLF on the morphology of the villus or the activity of digestive enzymes. Pigs are monogastric omnivores with a gastrointestinal anatomy very comparable to that of humans. Therefore, the use of the piglet model in those studies allows these findings to be extrapolated to humans. The inclusion of bLF in infant formula provides infants with a diet more similar to that of human milk, and may support intestinal development during the neonatal period [51]. As reviewed by Miyakawa et al. [52], several clinical studies have shown that bLF can improve child health, in particular by eliminating or relieving gastrointestinal and respiratory problems, and by improving the iron levels of children suffering from anemia, or those at high risk of anemia, as mentioned in the previous section.

BLF has been extensively studied, in comparison with hLF, as a substitute for the latter in some specialized products [53]. However, other alternatives, such as rhLF obtained in cow’s milk, have also been assayed regarding intestinal growth and maturation [37]. RhLF has shown a similar effect in promoting the establishment of the intestinal barrier than the bovine counterpart. It has also been postulated that it may be better to make bovine milk more similar to human milk. Nevertheless, the safety of transgenic foods remains controversial, and the widespread use of rhLF should be investigated more thoroughly [37] as the first attempts, in 2006 and 2010, for rhLF from rice to be recognized as GRAS by the FDA, were not successful [54].

The bioactivity of LF has also been studied in conjunction with other molecules, such as osteopontin (OPN). These two molecules are whey multi-functional proteins that are present at high levels in human milk. Since LF is a basic glycoprotein and OPN is an acidic phosphorylated glycoprotein, the two have high affinity for each other because of their opposite charges [55]. The LF–OPN complex showed a strong stability in in vitro digestion and more effective binding and uptake by human intestinal cells (HIEC) than either LF or OPN, alone. Furthermore, the LF–OPN complex promoted intestinal cell proliferation and differentiation to a greater extent than the individual proteins. Thus, it is hypothesized that when LF and OPN form a complex in human milk, they can protect each other against proteolysis, and potentiate their individual bioactivities [55]. The same effect was observed with bLF, as it can also form such a complex with OPN. The bLF–OPN complex was evaluated at different molar ratios (bLF:OPN = 3:1, 5:1 and 8:1) in a formula protein matrix that also included bovine whey protein hydrolysate and α-lactalbumin [56]. The bLF−OPN complex, together with formula proteins, was found to be resistant to in vitro digestion, and to stimulate intestinal cell proliferation (about 15−50%) and differentiation (about 30−50%). The 3:1 ratio of bLF to OPN exhibited the strongest effects, when compared with the other ratios [56]. As LF binds iron and OPN binds calcium, the binding of ligands may have an influence on the complex formation and the proliferation ability of the LF–OPN complex; therefore, the mechanisms underlying this effect were explored. BLF–OPN complexes were made and the effects on the proliferation of HIECs were evaluated [57]. Of the four complexes composed of apo- and holo-LF/OPN, the apo-LF–holo-OPN complex showed the greatest pro-proliferative activity on HIECs. This complex resisted gastrointestinal in vitro digestion; was co-localized with LF and OPN receptors, as confirmed by confocal microscopy; and promoted the proliferation of HIECs by triggering PI3K/Akt signaling [57]. Taking all this into account, the addition of bLF and OPN to infant formulas can enhance the stability of both components and improve their bioactivities, possibly improving the health of formula-fed infants.

### 3.2. Lactoferrin and Intestinal Damage Repair

LF also has protective effects on intestinal barrier function; to maintain the dynamic balance between the body and the intestinal tract, a complete barrier system is crucial. HLF has been reported to enhance the barrier function of a Caco-2 cell layer damaged by bacterial lipopolysaccharide (LPS) [58]. In this work, hLF showed an inhibitory effect on the decrease in cell viability caused by LPS. Moreover, the tight junctions of Caco-2 cell monolayers were evaluated by measuring the transepithelial electrical resistance (TEER) and the permeability of FITCT-labeled dextran 4000 (FD-4). In the presence of LPS, the preincubation of Caco-2 cells, with hLF at 400 and 1000 µg/mL, enhanced TEER significantly, compared to cells with no LF addition. Furthermore, at the same doses, the preincubation with LF suppressed the LPS-mediated increase in the permeability of Caco-2 monolayers. Similar findings were reached in other works. Thus, Zhao et al. [19] found that bLF reduced the paracellular permeability and enhanced alkaline phosphatase activity and transepithelial electrical resistance, reinforcing the barrier function of Caco-2 and human intestinal epithelial crypt cells (HIECs). In addition, bLF markedly upregulated the expression of the key tight junction proteins. These are claudin-1, occludin, and ZO-1, at the level of both the mRNA and protein, and consequently, reinforced the barrier function of both cell models. The research of Wang et al. [37] found that hLF, bLF, and rhLF from cow’s milk inhibited the reduction of Caco-2 cell viability caused by LPS. In addition, preincubation with hLF, bLF, and rhLF from cow’s milk, over 24 h, decreased the reduction in TEER and the increase in permeability to FITC-dextran of the cell monolayer induced by LPS.

The pig has gained importance, among the species that have been used as models for the study of human diseases, in recent years. In many regards, as it has been previously mentioned, the growing young pig is biologically closer to humans than the most widely used rodent models. In terms of the gastrointestinal tract, pigs and humans have similar digestive passage rates, digestive and absorptive capabilities, and intestinal length to body weight ratios, thereby making a good model for the study of gastrointestinal tract development [59]. A malnutrition pig model was chosen to evaluate the effect of bLF and rhLF from transgenic cows on the injured intestine. Weaning pigs were given a protein and calorie-restricted diet for five weeks, receiving daily cow milk, cow milk containing rhLF, or no milk supplementation from weeks 3 to 5. After 3 weeks, the restricted diet produced several changes in growth, blood chemistry, and intestinal structure (villous atrophy, increased ex vivo permeability, and decreased expression of tight junction proteins). An increase in growth rate, calcium and glucose levels, as well as intestinal epithelial growth, was observed when cow’s milk and rhLF were added to the diet. At the jejunum level, rhLF milk resulted in the restoration of the intestinal morphology, reduction of permeability, and the enhancement of anti-inflammatory IL-10 expression. Taking all this into account, LF can be a good candidate to promote the repair of damage in the intestinal epithelium caused by malnutrition [59].

LF was also suggested as an alternative treatment for an aflatoxin M1 (AFM1)-induced compromised intestinal barrier [60]. Regarding the results obtained using BALB/c mice and differentiated Caco-2 cells, the protective effects of bLF on the intestinal barrier dysfunction were related to two types of signaling pathways: one related to epithelial cell viability and the other with intestinal integrity. All the pathways associated with cell viability have been reported to be involved in the process by which LF stabilizes the intestinal barrier function. Those pathways included cell cycle, apoptosis, and mTOR, MAPK, p53, and PI3K-Akt signaling pathways [61,62,63]. In the pathways associated with the intestinal integrity the endocytosis, tight junction, adherens junction, gap junction, focal adhesion and ECM-receptor interaction are participating. Combining the analysis of transcriptome and proteome, a study by Gao et al. [60] proposed that insulin receptor (INSR), cytoplasmic FMR1 interacting protein 2 (CYFIP2), the dedicator of cytokinesis 1 (DOCK1), and ribonucleotide reductase regulatory subunit M2 (RRM2), were the potentially key regulators for the repair of the compromised intestinal barrier by LF, as all of them were previously identified as exerting a protective effect on the intestinal barrier [64,65].

These insights bring new data on the benefits of LF, which can be used as a functional food to repair intestinal epithelial lesions from various origins.

## 4. Lactoferrin and the Intestinal Immune System

LF plays an important role in modulating the development of the immune system. In fact, breastfed neonates have faster gastrointestinal and immune development than formula-fed infants [2]. Moreover, LF could be considered as a supplement for the prevention and treatment of diseases in which intestinal immune homeostasis is compromised [66]. This protein displays its effects on intestinal immunity by modulating both innate and adaptive immune responses.

### 4.1. Regulation of the Innate Immune System

LF exerts its effects on the innate immune system through cellular and molecular mechanisms. At the cellular level, LF can interact with different immune cells, as LFRs have been identified in lymphocytes, macrophages, and dendritic cells, triggering cellular responses such as migration, maturation, and proliferation [67]. It has been shown that LF promotes dendritic cell maturation, enhances natural killer (NK) cell activity, increases neutrophil migration, and induces macrophage activation, increasing their phagocytic capacity [68]. Iigo et al. [69] observed that patients with colorectal polyps, who ingested bLF, had an increased number of NK cells in their polyps. In addition, Spadaro et al. [70] showed that talactoferrin (TLF), an rhLF, increased Peyer’s patch cellularity, including NK cells, in transgenic BALB/c mice.

At the molecular level, LF interacts with a wide variety of compounds, both soluble and membrane molecules. At the surface of cells, some receptors that bind LF are CD14, low-density lipoprotein receptor-related protein-1 (LRP-1), intelectin-1 (omentin-1), and Toll-like receptors 2 and 4 (TLR-2, TLR-4) [71]. Soluble molecules can also bind to LF, such as lipid A of LPS or unmethylated CpG-containing oligonucleotides [72].

Accumulating studies, with animal and human models, report that LF can be considered a potent anti-inflammatory for the prevention and treatment of gastrointestinal inflammatory diseases [1]. This protein helps to limit excessive inflammatory responses by inhibiting the production of pro-inflammatory cytokines, such us TNF-α, IL-1β, IL-6 [73,74,75,76], and IL-8 [77,78], and by stimulating the synthesis of anti-inflammatory cytokines including IL-4 and IL-10 [74,75,76]. The reason for this effect of LF seems to be associated with different mechanisms of action. On the one hand, LF can modulate the inflammatory response by binding to lipopolysaccharide (LPS) on Gram-negative bacteria, or to the soluble (sCD14) or membrane (mCD14) receptor. This interferes with the formation of the LPS–CD14 complex, resulting in an attenuation of the Toll-like receptor 4 (TLR-4) signaling pathway [79,80]. The interaction of LPS with TLR-4 induces intracellular biochemical signaling that concludes with the release of pro-inflammatory cytokines, which, when produced in excess, can lead to tissue damage [81]. Therefore, through their LPS-binding properties, LF can counteract the LPS-induced inflammatory response, as well as the intestinal damage [73]. On the other hand, LF may act as a novel transcriptional regulator through LFRs located on the brush border membrane of the small intestine [82]. After LF binds to its receptor, it is internalized into intestinal cells by endocytosis and subsequently translocated into the nucleus. Inside the nucleus, LF binds to specific locations on DNA, thereby acting as a transcription factor, regulating the expression of various genes, including those involved in the inflammatory response [83]. Moreover, it has been shown that LF binds to bacterial DNA sequences (CpG motifs) in extracellular compartments, and thereby inhibits NF-kB activation and the subsequent expression of pro-inflammatory cytokines. One study investigated LF inhibition of CpG-induced NF-kB activation in B cells, and found that this protein strongly inhibited IL-8 and IL-12 transcription in B cells [84].

LF may also act as a potent anti-inflammatory agent by scavenging reactive oxygen species (ROS), which are strongly produced by granulocytes in the course of inflammation [85]. ROS induce inflammation through their ability to increase the production of IL-8 in intestinal epithelial cells. Therefore, an excess of these species enhances the inflammatory response, which contributes to the appearance of clinical symptoms [77]. LF is able to maintain the physiological balance of ROS levels by chelating free iron, which is essential for their production, or by regulating key antioxidant enzymes [86]. 

Moreover, LF seems to limit inflammation by inhibiting granulocyte migration. Cooper et al. investigated the effects of consuming transgenic milk, containing rhLF, on small intestinal eosinophils. They observed reduced eosinophil infiltration in the duodenum of piglets fed with rhLF-milk [87].

Although the inflammatory response is important in tissue repair and/or pathogen eradication, excessive and uncontrolled inflammation often leads to chronic diseases. In general, inflammatory bowel disease (IBD) patients exhibit an excessive immune response to commensal bacterial antigens, resulting in the release of pro-inflammatory cytokines, such as IL-1β, TNF-α, IL-6, and IL-12 [88]. Furthermore, the number of intestinal eosinophils has been shown to be elevated in ulcerative colitis and Crohn’s disease, indicating that these cells could be related to the pathology of these diseases [87]. In this regard, the anti-inflammatory effects of LF have been reported in different IBD animal models. Table 1 and Table 2 summarize the anti-inflammatory and immunomodulatory properties of LF in different in vivo and in vitro models. Taken together, these studies suggest that this protein could be considered a promising tool for the prevention and treatment of intestinal inflammatory disorders such as necrotizing enterocolitis or IBD. 

### 4.2. Regulation of the Adaptive Immune System

In addition to modulating innate immunity, LF also acts as a profound modulator of the adaptive immune system, bridging the two types of immune response. LF promotes the differentiation of immature T and B cells into competent T helper cells and antigen-presenting cells (APCs), respectively [68]. In murine models, oral administration of bLF has been shown to increase the number of CD4(+) and CD8(+) T cells in the lamina propria of the small intestine [89]. Moreover, recent studies have found evidence that LF modulates the balance between the Th1 and Th2 immune responses, commonly defined by the IFN-_Y_ and IL-4/IL-5 cytokines, respectively [90]. Indeed, LF enhances the ability of macrophages to function as APCs by stimulating the production of cytokines, such as IL-12, which is an important inducer of Th1 cell development [91]. In this regard, LF can modulate Th1/Th2 cell activities, depending on the host immune status. For example, LF promotes Th1 responses in diseases that require a strong cellular response (infectious diseases or cancer), but LF can also reduce the Th1 component to limit excessive inflammation [92].

As for B cell differentiation, LF administered to mice increased the number of IgA (+), IgM (+) [89], and IgG (+) B cells [93,94] in the lamina propria or Peyer’s patches of the small intestine. This research has clearly shown that orally administered LF enhanced total IgA production in the intestinal mucosa. Moreover, other studies, using mouse models infected with Salmonella typhimurium, showed that LF treatment further enhanced the secretion of *S. typhimurium*-specific IgA. These results suggest that LF has an important effect on the mucosal/systemic IgA response and can contribute to protection against intestinal pathogens [95,96].

**Table 1 pharmaceutics-15-01569-t001:** Anti-inflammatory and immunomodulatory properties of LF in different in vivo models. Increase (↑), decrease (↓).

In Vivo Models	LF Source/Dose/Time	LF Activity	Reference
Patients with adenomatous colorectal polyps	1.5 or 3 g bLF daily for 1 year	↑ IFN-α in the colon ↑ NK cell activity and CD4(+) cells in the polyps	[69,97]
Healthy humans	100 mg bLF for 7 days, followed by 200 mg bLF for 7 days	↑ CD4(+) and CD8(+) T cells in peripheral blood lymphocytes	[98]
Very-low-birth-weight neonates	200 mg bLF daily throughout hospitalization	↑ Treg levels	[99]
BALB/c mice with implanted tumours	1000 mg/kg talactoferrin (TLF) daily for 3 weeks	↑ IFN-_Y_ production in intestinal mucosa ↑ CD8(+) T lymphocytes and NK cells in Peyer´s patch	[70]
Healthy C57BL/6 mice	0.2% and 2% bLF for 4 consecutive days	↑ Apoptotic CD4 and ↓ TNF-α expression in intestinal lymphocytes	[100]
Healthy BALB/c mice	30 mg/kg/day of bLF for 7 days or a single administration at 300 mg/kg on day 7	↑ Mature IL-18 in the mucosa of the small intestine	[101]
BALB/c mice stressed by immobilization	bLF (50, 500, and 5000 μg) for 7 days	↑ Total IgA, secretory IgA, IL-4, and IL-6 (500 µg) in proximal intestine ↑ Total IgA, IL-4 (5000 µg) in distal intestine	[102]
BALB/c mice infected with a lethal or sublethal dose of S. *typhimurium*	5 or 100 mg of bLF for 7 days	↑ Total and S. *typhimurium*-specific IgG, and IgM in serum and IgA in intestinal secretions	[95]
Healthy BALB/c mice	5 mg of bLF for 7, 14, 21, or 28 days	↑ IgA and IgM antibodies, IgA+ and IgM+ plasma cells, total B cells, CD4(+) and CD8(+) T lymphocytes either in Peyer’s patches or lamina propria of the distal small intestine	[89]
Healthy BALB/c mice	0.05 mg/g or 1 mg/g body weight of bLF for 4 weeks	↑ Total immunoglobulins, IgA, and IgG in the intestinal fluid and in Peyer’s patches	[94]
BALB/c mice infected with an avirulent strain of S. *typhimurium*	bLF (5 mg/Kg) three times a week for 3 weeks	↑ S. *typhimurium*-specific IgA, IgG2b, and IgG1 in fecal pellets and sera	[96]
Healthy BALB/c mice	bLF (50, 500, or 5000 μg) for 7 days	↑ Total and specific IgA in intestinal secretions ↑ IL-2 and IL-5 in intestinal mucosa	[103]
Vitamin D-deficient C57BL/6J mice	100 mg/kg and 1000 mg/kg of bLF for 24 weeks	↓ Serum inflammatory cytokines (TNF-α, IL-6, and TGF-β) ↓ TLR-4 expression and NF-κB P65 activity in the colon	[104]
Specific-pathogen-free BALB/cByJ Jcl mice	bLF (2.5 g/kg of body weight) for 1 day	↓ TLR3 gene, IFN-γ and IL-10 and ↑ NOD2 gene, IFN-β, and IL-12p40 in the small intestine	[105]
Specific pathogen-free C57BL/6 mice	bLF (30, 100, 300, 1000 mg/kg body weight) for 7 days	↑ IL-18, IFN-α, IFN-β, and NK cell activity in Peyer’s patches and mesenteric lymph nodes	[106]
C57BL/6 mice with intestinal dysbiosis induced by clindamycin	35 mg of native or iron-saturated bLF for 10 days	Saturated bLF reverted the decrease in the expression of TLR2, TLR8, and TLR9 induced by clindamycin in the colon	[107]
Healthy C57BL/6 mice	500 mg/kg bLF for 3 days	↑ IFN-_Y_ and IL-10 production by intestinal intraepithelial lymphocytes (IEL) and mesenteric lymph-node (MLN) cells	[108]
DSS murine colitis model and TNFΔ^ARE/+^ murine model of ileitis	500 mg/kg/day of a rhLF (VEN-120) for 7 or 14 days	↑ Treg levels in the intestinal lamina propria ↓ IL-17 and IFNγ, and ↑ IL-10 produced by CD4+ T cells of the lamina propria	[109]
C57BL/6 mice with DSS-induced colitis	2 mg/mouse of hLF twice a day during DSS exposure	↓ Serum IL-1β levels and IL-12 ↓ CD4 cells, F4/80+macrophages, and TNF-α producing cells in the distal colon	[110]
Mouse model with intestinal damage induced by LPS	100 mg/kg/day of holo-rhLF or apo-rhLFfor 14 days	Apo-rhLF ↓ TNF-α, IL-1β, and IL-6 in the colon and ↑ Expression of anti-inflammatory factor, IFN-γ	[73]
BALB/c mice with DSS-induced colitis	Diet containing 1.25% (wt/vol) of bovine iron-saturated LF (Fe-bLF)	Fe-bLF ↓ Myeloperoxidase activity (neutrophil infiltration) ↓ DSS-induced expression of IFN-γ, TNF-α, IL-4, IL-5, GM-CSF, and NO in the colon	[111]
Mouse model of DSS-induced colitis	100 mg/kg of bLF for 14 days	↓ IL-1β, IL-6, and TNF-α in the colon ↑ IL-10 and TGF-β in the colon	[76]
C57BL/6 mice with necrotizing enterocolitis	rhLF (0.3 g/kg/day) for 3 days	↓ IL-6 and TNF-α expression, and ↑ intestinal stem cell marker Lgr5	[112]
Rats with DSS-induced colitis	200 mg/kg/day of bLF starting 3 days before beginning DSS administration, until death.	↓ Pro-inflammatory cytokines TNF-α, IL-1β and IL-6 in the colon ↓ Myeloperoxidase activity ↑ Anti-inflammatory cytokines IL-4 and IL-10 in the colon	[75]
A preterm pig model	10 g/L of bLF-enriched formula	bLF ↓ IL-1β in the proximal small intestine	[113]
Healthy piglets	2, 11, or 20 mg/g of rhLF for 30 days	Enhanced Th1 (↑ IL-12) and Th2 (↑ IL-10) cell responses ↑ TLR2 and NF-κB P65 in the intestine ↑ Total IgG and IgA in the plasma and ↑ IgG in the colon	[114]

**Table 2 pharmaceutics-15-01569-t002:** Anti-inflammatory and immunomodulatory properties of LF in different in vitro models. Increase (↑), decrease (↓).

In Vitro Models	LF Source/Dose/Time	LF Activity	Reference
Caco-2 cells	50 mg/mL of TLF, bLF or hLF for 72 h	hLF ↑ expression of TGF-β1	[115]
Caco-2 cells	Native and Fe-saturated bLF and hLF (400 mg/mL or 50 mg/mL) for 72 h	Native forms ↑ TGF-β1 Holo-forms ↑ IL-18 secretion	[12]
Caco-2/TC7 cells	bLF (0.5, 1, 2, 5 or 10 mg/mL) for 24 h	↓ Expression levels of TLR4 with 10 mg/mL	[116]
Caco-2/TC7 cells	100 μg/mL bLF for 3 h	↑ IFN-α, IFN-β, TLR-3, TLR-7, IRF3, and IRF7	[117]
SARS-CoV-2 infected-Caco-2 cells		LF pre-infection treatment ↓ IL-1β, IL-6, and IL-10, and ↑ TGF-β1	
Caco-2 and RAW 246.7 cell lines treated with LPS	1, 1.5 and 2 mg/mL of rhLF	↓ IL-8 and ROS production	[77]
Caco-2 cells and organ cultures infected with *E. coli* strain LF82	1 mg/mL bLF	↓ TNF-α, IL-8, and IL-6	[78]
Rat intestinal epithelial cells IEC-18 and Caco-2 cells treated with H_2_O_2_	0.01, 0.1 and 1 g/L of rhLF for 24 h	↓ IL-6 and ↑ gene expression of Lgr5 and Wnt/β-catenin	[118]
Peritoneal macrophage culture	1000 mg/mL of bLF for 24 h	↑ Mature IL-18, IFN-_Y_ and IL-15 and ↓ expression of IFN-α	[101]
RPMI 8226 B cells	0.5–4 µmol/L apo-LF for 1 h before stimulation with CpG for 16 h	↓ NF-kB activation ↓ IL-8 and IL-12p40 expression induced by CpG	[84]
Porcine intestinal epithelial cells (PsIc1) treated with LPS	bLF (0–10 g/L) for 24 h	Low doses (0.1–1 g/L) ↓ IL-8, and NF-κB and hypoxia-inducible factor-1α (HIF-1α) activation	[113]
Intestinal epithelial cell line HT-29	bLF (0.002, 0.02, 0.2, or 2 mg/mL) for 1 h prior to the addition of poly I:C	↑ IFN-λ1 in the culture supernatant and ↑ IFN-λ1 and IFN-λ2 mRNA levels	[119]
LPS-challenged THP-1 cells	0.125–2 mg/mL bLF	↓ TNF-α release from LPS-activated THP-1 cells	[120]

### 4.3. Lactoferrin as Fecal Biomarker of Inflammation

During intestinal inflammation, polymorphonuclear neutrophils infiltrate the mucosa and release proteins and cytokines [121]. Among the proteins secreted by neutrophils, we can find LF, which is the main component of neutrophil secondary granules. This protein is released by degranulation during an inflammatory process, resulting in increased levels of LF in feces, and its concentration is proportional to neutrophil translocation to the digestive tract [122,123]. Consequently, as with calprotectin, this protein can be used as a fecal biomarker of intestinal inflammation. In fact, LF has key characteristics desirable for clinical use. Firstly, this protein is resistant to proteolysis and is not affected by several freeze/thaw cycles. In addition, it has been reported that LF can remain stable in feces at room temperature for 5 days [124]. Secondly, fecal LF can be easily detected and quantified using ELISA and immunochromatography [123].

Fecal markers of IBD have been investigated by many authors, for diagnostic purposes. Their use holds great promise because they are non-invasive, cheap, easy to sample, and therefore, they can be re-sampled more frequently. Furthermore, compared to serum biomarkers, they are more sensitive to intestinal inflammation because they have direct contact with inflamed tissues [124,125]. Among these fecal markers, LF and calprotectin have proven their value in detecting active IBD, predicting disease recurrence, or assessing responses to medical treatment [123]. Although calprotectin is ordered much more frequently, studies evaluating LF in the diagnosis of IBD show that calprotectin and LF results have high concordance [126,127]. In fact, specific antibodies against both proteins are used together in some commercial tests to detect IBD (www.certest.es). Nevertheless, fecal LF measurements would have limited utility in breastfed infants, due to the presence of LF from breast milk [124].

Elevated fecal LF levels have been found in patients with ulcerative colitis and Crohn’s disease, with sensitivities ranging from 78–90% and specificities from 90–100% [123,128,129]. In addition, LF concentrations show significant correlation with endoscopic assessment of intestinal injury; therefore, patients with more severe endoscopic disease activity have higher LF concentrations in feces [130,131]. This biomarker is also useful to screen for inflammation associated with IBD versus irritable bowel syndrome (IBS). IBS is a functional non-inflammatory disorder with similar symptoms to IBD; thus, an elevated fecal LF result is 100% specific to ruling out IBS [126,129].

Regarding infectious diarrhea, some authors have suggested the use of fecal LF as an important tool to predict and monitor the clinical severity and course of gastrointestinal infections. In Cryptosporidium parvum-infected children, growth deficits were associated with the degree of oocyst excretion and LF positivity [132]. Moreover, increased levels of fecal LF were linked to a greater number of days and episodes of diarrhea in patients with Giardia duodenalis infection [133]. Chen et al. [134] found that concentrations of fecal LF increased during bacterial infection (Salmonella and Campylobacter) and with higher severity of illness, compared to patients with viral infections and mild disease activity.

## 5. Lactoferrin in Some Intestinal Disorders Caused by Bacteria

### 5.1. Lactoferrin’s Effect on Late-Onset Sepsis and Necrotizing Enterocolitis

Late-onset sepsis (LOS) and necrotizing enterocolitis (NEC) are two severe illnesses affecting low-birth-weight infants (LBWIs) and newborns. NEC is characterized by a hemorrhagic and necrotizing inflammation affecting all layers of the intestinal wall [135]. Although the pathophysiology of NEC has not yet been established, it seems that the translocation of gas-producing bacteria, from the lumen of the GIT into the intestinal wall, significantly influences NEC pathophysiology. The role of human milk in both prevention and treatment of NEC has been accepted and the administration of donor milk is one of the treatments recommended by pediatricians. Enteral feeding has been frequently associated with NEC onset. This type of feeding causes a low diverse intestinal microbiota with a predominance of pathogens, which produces inflammation of the intestinal tissues, with a decrease in blood flow and coagulation and necrosis of the bowel [136]. Human milk provides significant protection against many of the factors that are recognized as promoters of NEC, as well as a therapeutic option for infant recovery from it, as has been indicated above. However, because human milk is scarce and some NEC cases are very severe, there have been intensive attempts to search for compounds that may improve the health status of infants with NEC, LF being one of them.

In a study by Manzoni et al. [137] a prospective, multicenter, double-blind, placebo-controlled, randomized trial was conducted on 472 patients that were LBWIs. They were divided in three groups: one group was administered with bLF (100 mg/day), the second group was administered with the same amount of bLF plus Lactobacillus rhamnosus GG (LGG, 6 × 10^9^ colony-forming units/day), and the third group with a placebo. The treatment was applied from birth to 30 days of life, and the results demonstrated that supplementation with bLF, either alone or in combination with LGG, reduced the first episodes of LOS and NEC incidence. Although it is not clear that LF can reduce the appearance of NEC and its symptoms, LF exerts anti-inflammatory actions that may mitigate the pro-inflammatory state that is present in the gut before the onset of NEC. Therefore, LF could be useful for prevention in LBWIs that cannot be breastfed by their mothers [136].

Turin et al. [138] reviewed the recent and ongoing clinical trials, on the use of LF for the prevention of neonatal sepsis, and found 11 registered clinical trials, including more than 6000 subjects. The preliminary results suggested a positive protective effect of LF on neonatal infections. One year later, Ochoa et al. [139] conducted a pilot randomized, placebo-controlled, double-blind study in Peruvian infants with a birth weight of <2500 g, which were randomized to receive bLF (200 mg/kg/day) or a placebo for four weeks. The results of the study showed that overall sepsis was less common in the LF group than in the control group, although the primary results were without statistical significance.

In the systematic review published by Pammi and Gautham [140], it was concluded that the results, published about the effects of LF in the prevention of sepsis and NEC in preterm infants, showed evidence of medium to low quality, suggesting that oral LF prophylaxis, with or without probiotics, decreases LOS and NEC (stage II or greater) without adverse effects.

More recently, two meta-analyses [141,142] concluded that the results of the studies, in which enteral supplementation with bLF was administered to very preterm infants, do not support its routine use to prevent morbidity and mortality of late-onset infection.

### 5.2. Lactoferrin against Some Pathogenic Bacteria Causing Intestinal Diseases

Although the etiology of inflammatory bowel diseases (IBD), such as Crohn’s and ulcerative colitis, remains unknown, there is evidence that an abnormal immune response against the microorganisms of intestinal microbiota can be one of the causes, the adaptive and innate immune responses being involved [143]. Moreover, some pathogenic bacteria, fungi, viruses, and parasites have been suggested to be involved in the development and exacerbation of IBD [144]. The bacteria most frequently associated with IBD are Mycobacterium avium subspecies paratuberculosis, *Clostridium difficile*, *Listeria monocytogenes*, and *Escherichia coli*, among others [145]. In particular, a pathovar of *E. coli* called AIEC (adherent-invasive *E. coli*) has been strongly associated with many Crohn’s diseases [146]. This pathogen is able to adhere to intestinal cells, to invade them and to replicate in epithelial cells and macrophages, eventually producing intestinal diseases in humans [146]. It was reported by Bertuccini et al. [78] that bLF inhibited AIEC invasion of the intestine, through minimally affecting adhesion. These authors hypothesized that this was due to the binding of LF to the bacterial type 1 pili, through its mannose residues. This binding caused the aggregation of bacteria to the intestinal cell surface, avoiding their internalization. Furthermore, the expression of pro-inflammatory cytokines, such as TNF-α, IL-8, and IL-6, was observed to be significantly suppressed by LF, both in the cell line Caco-2 and in Crohn-derived intestinal cells from biopsies, when both cells were infected with AIEC. Afterwards, Lepanto et al. [147] reported that bLF pre-treatment of differentiated Caco-2 cells activated protective intracellular pathways, thus reducing the invasion of cells by bacteria and their survival, and consequently, avoiding damage to cell DNA.

There are other pathotypes of *E. coli* which cause enteric diseases, such as enteropathogenic *E. coli* (EPEC), enterohemorrhagic *E. coli* (EHEC), enterotoxigenic *E. coli* (ETEC), enteroaggregative *E. coli* (EAEC), enteroinvasive *E. coli* (EIEC), and diffusely adherent *E. coli* (DAEC) [148]. A study by Ochoa et al. [149] was conducted to evaluate the effects of bLF on the adherence of EAEC to Hep-2 cells. The results showed that LF, with or without iron, inhibited the adherence of bacteria to cells and the formation of biofilms, independently of the iron content. The authors hypothesized that the effects of LF are due to its interaction with key adherence proteins, such as aggregative adherence fimbria (AAF) or dispersin.

EHEC is a foodborne pathogen that can cause mild diarrhea and hemorrhagic colitis and, in some cases, hemolytic uremic syndrome. All these symptoms are caused by the Shiga toxins produced by EHEC. In search of alternative compounds to antibiotics, it was conducted an assay in a murine model, orally infected with EHEC, to evaluate the activity of bLF and LFchimera (consisting of lactoferricin and lactoferrampin, two known peptides derived from bLF). The results showed that bLF and LFchimera decreased mortality, sepsis, number of bacteria in feces, and their presence in the liver and kidney [150]. Afterwards, Rybarczyk at al. [151] demonstrated, in the in vitro Vero cell model, that bLF reduced the cytotoxic effect of two different strains of EHEC, producers of Shiga toxins 1 and 2. This effect could be due in part to the degradation of the Shiga toxin 2 receptor-binding B-subunit by LF. This finding could support the use of LF as alternative treatment in infections by EHEC, as the Shiga toxin 2 is the toxin most frequently isolated in human patients.

The pathotype, EPEC, has been reported as a significant cause of infant diarrhea in developing areas. This bacterium causes a characteristic histopathological lesion in the enterocytes, called ‘attaching and effacing’, that produces cytoskeletal changes, mediated by the type-III secretory system [150]. In a study by de Araújo and Giugliano [152], the activity of the non-immunoglobulin fraction of human milk, which inhibits adhesion of EPEC to Hela cells (with a similar pattern to enterocytes), was evaluated. They found that the free secretory component and LF were able to inhibit the adhesion of EPEC to the cells, even at concentrations lower than those normally present in human milk. In a cellular model with Hep-2 cells, it was observed that bLF and LFchimera could inhibit the first steps of the attaching process of EPEC to these cells, and that this effect can be related to the type-III secretory system [150]. The authors also hypothesized that LFchimera could interact with bacterial DNA, interfering with the expression of some virulence factors.

With respect to ETEC, Kawasaki et al. [153] conducted an experimental assay on human epithelial cells, in vitro and on the intestinal mucosa of germ-free mice in vivo. In the in vitro assay, they found a positive inhibitory effect of bLF against the adherence of ETEC to cells. Likewise, in the in vivo assay, they found significantly lower counts of bacteria in several segments of the intestine in the group treated with LF, thus suggesting a blocking effect of LF against ETEC adherence to intestinal mucosa.

*C. difficile*, in addition to being involved in IBD, has generated significant concern in recent years because it causes severe colitis, whose treatment is difficult, due to its resistance to antibiotics [154]. *C. difficile* is the main cause of diarrhea treated in hospitals, with high morbidity and the consequent economic burden for the national health system of governments [155]. *C. difficile* mainly produces toxin A and toxin B, which cause intestinal mucosal damage. Great efforts have been made to find alternative compounds to substitute antibiotics in the treatment of *C. difficile* infection, to avoid the problem of resistances. As it has been proven that *C. difficile* requires iron for its growth, research on the capacity of LF to fight against this infection has been conducted in recent decades. Thus, Chilton et al. [155] evaluated the effects of apo-LF and holo-LF on *C. difficile* spore germination and growth of vegetative cells in a triple stage of a chemostat gut model. Contrary to what might be expected, the most effective form of LF was holo-LF, as the level of spore germination and the production of toxins was lower than that obtained with apo-LF. The hypothesis of the authors was that holo-bLF could be more resistant to degradation and proteolysis, and more bioavailable than apo-bLF. Another possibility is that iron binding to LF induces conformational changes in the molecule that facilitates the interaction with *C. difficile*’s cell wall.

In another study performed in vitro, the effect of LF and pepsin-treated LF was assayed on a rat intestinal epithelial cell line (IEC-6), after challenge with *C. difficile* toxin B, which is more cytotoxic for these cells than toxin A. The results showed that LF and pepsin-treated LF prevented the toxin B-induced cytotoxicity, by reverting the damage—mainly on the tight junction’s structures—and avoiding the reduction in cell proliferation, the whole LF molecule being more protective than the pepsin-treated LF [156].

In this sense, a study by Braim et al. [157] showed that chitosan-coated alginate microparticles, loaded with bLF, exerted a protective effect against *C. difficile* bacteria, and the damage of the barrier function in Caco-2 cells (differentiated in enterocytes) and Vero cells (monkey kidney epithelial cells) produced by its toxin.

The rapid diagnosis of *C. difficile* infection is essential in order to provide an adequate treatment for patients, and to avoid transmission to others. As it has been reported, many patients positive for *C. difficile* infection do not present clinically significant diarrhea. Therefore, there is a need for specific biomarkers of this infection. Although it has been considered that LF could be a good biomarker to detect *C. difficile* infection, no clear conclusions have been drawn until now [158].

### 5.3. Lactoferrin against Some Viruses Causing Intestinal Disorders

The most common viruses causing intestinal disorders are rotavirus [159] and norovirus [160]. The antiviral activity of bLF against the simian rotavirus, SA-11, was evaluated in the human colon adenocarcinoma cell line HT-29 by Superti et al. [161]. They evaluated the activity of LF after saturation with different metals. The results showed that the antiviral activity of LF, fully saturated with manganese or zinc, diminished to some extent, compared to that observed for apo- or iron-saturated LF. The antiviral activity of differently metal-saturated LF towards rotavirus was exerted during and after the virus attachment step. They also evaluated the activity of LF after removal of sialic acid, which, surprisingly, improved the anti-rotavirus activity of LF. Furthermore, when studying the effect of LF hydrolysis with trypsin, on its antiviral activity, they found that a large fragment (86–258) and a small peptide (324–329: YLTTLK) were able to inhibit rotavirus, although to a lower extent than undigested LF. 

In a study by Parrón et al. [162], the antirotaviral activity of LF, from the milk of various species, was evaluated in in vitro assays on an MA104 cell model. The results showed that bovine, ovine, swine, camel, and recombinant human LFs (at 1 mg/mL) neutralized the infection of MA104 cells by bovine rotavirus WC3 strain, with neutralization values in the range of 88–99%. However, LF from human milk only showed neutralizing activity of around 58%. Iron saturation of bLF did not modify its antirotaviral activity. In the same study, the effect of technological treatments on LF antirotaviral activity was also evaluated, and the results revealed interspecies differences in LF heat susceptibility. Pasteurization at 75 °C for 20 s preserved LF activity very well, and the non-thermal treatment of high pressure at 600 MPa for 15 min did not cause any significant loss of the activity of LF.

In a study by Zavaleta et al. [163], it was shown that the addition of rhLF and lysozyme to a rice-based oral rehydration solution had a beneficial effect on children with acute diarrhea, though the effect was not analyzed with respect to the causal agent, and there was also more than one pathogen identified in the stool samples of infants. However, in a trial conducted by Yen et al. in 2011 [80], to determine whether a formula containing bLF exerted a protective effect against the enterovirus, EV71, or against rotavirus infection among children from 2 to 6 years old, the results were not so promising. Actually, an oral supplement of LF, at a dose of 70 mg/day, did not show any benefits in the prevention of EV71 or rotavirus infection, or demonstrate any impact on the IFN-γ or IL-10 serum levels in healthy children participating in the trial.

Human noroviruses are non-enveloped, single-stranded DNA, and cause the majority of acute gastroenteritis in humans, with higher morbidity and mortality in infants, the elderly, and immunocompromised individuals. Unfortunately, there are no treatments or effective vaccines available against noroviruses [164]. Therefore, it is very important to find compounds that can alleviate the symptoms produced by norovirus infection. Due to the initial difficulty of growing human norovirus in in vitro cellular or in vivo models, Ishikawa et al. [164] evaluated the activity of bLF against a murine norovirus in a murine cell line. They found that LF inhibited the attachment of the noroviruses to the cells, their replication, and the enhancement of antiviral immunity. Later, Oda et al. [165] were able to grow human norovirus in a B cell culture model, and found that bLF reduced human norovirus infection, possibly through an indirect mechanism involving the induction of innate interferon responses.

## 6. Modulatory Effects of Lactoferrin on Gut Microbiota

The intestinal microbiome plays an important role in the protection of intestinal health, and several factors such as antibiotics, infections, and stress will cause dysbiosis and different pathologies in the gut. LF is an iron-containing glycoprotein that is able to compete with pathogens for iron ions and generate electrostatic attraction with bacterial cell membrane components, contributing to the modulation of gut microbiota [80,166]. The efficacy of LF as a selective modulator of the microbiome has been established in several animal models and human studies.

### 6.1. Mouse Models

The oral administration of bLF (100–1000 mg/kg) in vitamin D-deficient mice reduces the abundance of Oscillibacter but increases the proportion of Lachnospiraceae, Faecalibaculum, and Lactobacillus in feces [104]. Oscillibacter is a genus of Gram-negative bacteria associated with intestinal inflammation [167]. Lachnospiraceae improves colonic inflammatory symptoms, reducing the expression of pro-inflammatory cytokines (TNF, IL-6, IL-1, IL-12) and inhibiting the NF-ΚB signaling pathway in dextran sulphate sodium (DSS)-treated mice [168]. Faecalibaculum reduces the inflammatory symptoms, by inhibiting IL-8 expression and activating NF-ΚB in TNBS-induced colitis [169]. Lactobacillus protects from intestinal inflammation by inhibiting NF-ΚB signaling pathway activation, which in turn improves elevated pro-inflammatory cytokines in vitamin D-deficient mice [170,171].

The supplementation of diet with bLF (100 mg/kg) suppresses high-fat-diet-induced obesity in mice by modulating the levels of *Bifidobacterium* species in gut microbiota [172]. *Bifidobacterium* spp. is a well-known probiotic with multiple beneficial effects on intestinal physiology, i.e., modulating gut microbiota, improving the immune system [173], or preventing intestinal diseases [174]. In addition, *Bifidobacterium* spp. enhance intestinal barrier function, promote a healthier microvillus environment, and reduce bacterial translocation [175], eventually ameliorating chronic inflammation. In line with these effects, bLF is able to restore the abundance of *Bifidobacterium* spp., which could be associated with reduced intestinal inflammation and systemic LPS level, and an anti-obesity effect [172].

The oral administration of bLf (100 mg/kg) protects against inflammation and damage to the colonic epithelial barrier, induced by DSS in mice, by increasing the proportion of bacteria from the Akkermansiaceae family [76]. *Akkermansia* spp. are mucin-degrading bacteria essential for preserving the mucus layer’s integrity, thus decreasing intestinal permeability and LPS leakage [172].

The supplementation of diet with bLF (100–1000 mg/kg) prevents ethanol-induced liver injury in mice by promoting Lactobacillus and *Akkermansia* richness [176]. In fact, supplements of Lactobacillus or *Akkermansia* can assist in improving alcoholic-associated liver disease. *Akkermansia* muciniphila protects the host against ethanol-induced gut leakiness, increased mucus thickness and tight junction expression, and improved ethanol-induced hepatic injury and neutrophil infiltration in mice [177]. Lactobacillus rhamnosus GG treatment promotes intestinal hypoxia-inducible factor, increased intestinal integrity, and improves alcohol-induced liver injuries in mice [178].

The administration of native and iron-saturated bLF (35 mg/day) positively modulates the intestinal microbiota of mice with intestinal dysbiosis induced by clindamycin, as the levels of Bacteroidaceae, Prevotellaceae, and Rikenellaceae are increased [107]. The Bacteroidaceae family includes many significant opportunistic pathogens, but as indispensable members of a stable microbiota, they are considered health-maintaining, as they boost the intestinal epithelial barrier and ameliorate inflammation by producing anti-inflammatory molecules such as polysaccharide A [179]. *Prevotella* spp. can improve glucose metabolism stimulated by the ingestion of prebiotics [180]. Members of the Rikenellaceae family are hydrogen-producing bacteria that selectively counteract cytotoxic reactive oxygen species and protect cells from oxidative stress [181]. In the inflammation process, H_2_ facilitates the suppression of pro-inflammatory cytokines in inflamed tissues [182].

The administration of LF (2%) in drinking water regulates metabolic disorders in nutritionally obese mice by regulating intestinal microbiota [183]. Obese mice treated with LF showed a decreased Firmicutes/Bacteroidetes ratio; a reduction in the levels of Deferribacteres, Oscillibacter, Butyricicoccus, Acinetobacter and Mucispirillum; and an increase in the abundance of Dubosiella. The ratio of Firmicutes to Bacteroidetes is generally considered as a biological marker for obesity [184]. The reduction of this ratio can control anomalous short-chain fatty acid metabolism and prevent a chronic, mild inflammatory response [185]. High levels of Deferribacteres are often related to obesity, which is positively linked with the pro-inflammatory factors, IL-6, IL-17A, and TNF-α, causing the aggravation of inflammation in obesity [186]. Dubosiella is a patented probiotic for modulating weight loss and preventing metabolism and immunity-associated diseases, such as obesity, diabetes, metabolic syndrome, and abnormal lipid metabolism [187]. Oscillibacter, Butyricicoccus, Acinetobacter and Mucispirillum are commensal bacteria that can induce intestinal inflammation or pathology in diet-induced obesity, diabetes, or any change in the gut environment [188].

### 6.2. Piglet Models

An extensive review of the modulation of the microbiota by LF in piglet models has been previously published [189], so only the most recent studies are highlighted in this review. It has been widely reported that supplementing the diet of piglets with bLF can promote the growth of beneficial microbes such as Lactobacillus and *Bifidobacterium* [190,191,192,193]. Lactobacillus, a predominant genus in the small intestine, improves nutrient absorption, alleviates inflammatory responses by reducing the expression level of TNF-α in a rat colitis model [194], prevents the infection or colonisation of pathogens by competing for epithelial binding sites and nutrients, and also produces antimicrobial factors, such as bacteriocins and lactic acid [195].

Early life LF intervention in suckling piglets, increases the abundance of Bacteroidetes and decreases the levels of Proteobacteria and Fusobacteria. The bacteroidete phylum contains many SCFA-producing bacteria, such as acetic-producing or butyrate-producing *Prevotella* and *Bacteroides*, which preserve intestinal health [196,197]. The Fusobacteria phylum, which includes the Fusobacterium genus, is closely related to human colon cancer, ulcerative colitis, and other diseases such as lameness and facial skin necrosis in piglets [198,199]. The Proteobacteria phylum includes various pathogens, such as Escherichia-Shigella, Salmonella, and Helicobacter. Escherichia-Shigella impairs the intestinal electrolyte balance and decreases the absorption of fluids, conditions that lead to intestinal dysfunction. It has been reported by several authors that the levels of these iron-dependent, Gram-negative bacteria are reduced in the intestines of piglets and/or gilts after supplementing diet with LF [190,191,192]. In addition, the levels of Gram-positive pathogenic bacteria of the Streptococcus genus have also been positively modulated by a diet containing LF [193].

The feces of piglets infected by porcine epidemic diarrhea virus (PEDV) have an abundance of Leptotrichia—which correlates with diarrhea—and Actinobacillus and Veillonella, which may cause dysfunction in energy and amino acid metabolism in the small intestine. The administration of LF in sucking piglets reduces the abundance of Leptotrichia, Actinobacillus, and Veillonella, suggesting that LF can help hosts maintain intestinal function by reducing the levels of these bacteria [192,193]

### 6.3. Human Studies

A few studies support the role of LF in the modulation of human intestinal microbiota. Talactoferrin (TLF), a drug designation for rhLF, modulates fecal Enterobacteriaceae in the feces of very-low-birth-weight infants. Enteral TLF prophylaxis lowered the burden of Enterobacter hormaechei, providing a possible mechanism for a reduction in necrotizing enterocolitis, as this bacterium heightens antimicrobial resistance, colonizes neonatal feeding tubes and causes bacteremia in neonatal intensive care units [200]. Progel microencapsulated bLF (Inferrin^TM^, Bega Bionutrients, Victoria, Australia) reduced the levels of Euryarchaeota and Chloroflexi, both bacteria that are associated with oral infections, especially periodontitis [201]. However, treatment, of persons living with human immunodeficiency virus (HIV), with rhLF did not result in significant changes to inflammation or immune activation within blood, or to the intestinal microbiota [202].

## 7. Conclusions

The first organ that receives LF, when it is ingested as a natural compound in milk or added as an ingredient in certain functional food products, is the gastrointestinal tract. In vitro studies have demonstrated that LF is not fully degraded in the stomach and the small intestine, allowing it to bind to brush border membrane receptors. These receptors mediate the specific uptake of LF into enterocytes and crypt cells, being transported from the intestinal lumen to the bloodstream. Therefore, LF can act not only at the luminal intestinal level but also in different tissues and organs. Several studies have indicated the nutritional significance of LF in regulating iron absorption in infants, and also as a bioavailable source of iron, equivalent in bioavailability to ferrous sulphate, in adult females. In fact, LF supplementation has been found to be effective in the treatment of iron deficiency pathologies. Oral administration of LF regulates iron homeostasis in humans by promoting iron absorption and by suppressing the inflammatory processes causing anemia in chronic diseases. Furthermore, it has been reported that LF oral administration did not produce any side effect, as occurs with other iron supplements.

The content of LF is very high in human colostrum and its concentration is also maintained at very good levels in mature milk. LF has been suggested to be a modulator of intestinal development through direct effects on intestinal epithelial cell proliferation and differentiation during infancy. Due to the mild gastric conditions in infants, the hydrolysis of LF is minimal and, therefore, LF might have greater biological potential in infants than in adults. It has been proposed that the high LF concentration in colostrum assists in the formation of the small intestine tissue mass, while, gradually, after a transition period, the main role of LF present in mature milk is to differentiate progenitor cells, to one of the functional cell types of the small intestine, and place them in their proper position. The relevant role of LF, in promoting the growth and maturation of the intestinal mucosa, suggests that it could be a therapeutic agent in premature infants, and also in patients with intestinal diseases such as Crohn’s disease. There has been much evidence of the potential beneficial properties of LF in intestinal damage repair, suggesting its use as a functional compound to repair intestinal epithelial injury due to different causes.

Furthermore, a complex, formed by LF and osteopontin in human milk, with a possible biological significance, has been found. This complex showed a strong stability against in vitro digestion and higher binding and uptake by human intestinal cells than LF or osteopontin by themselves. Moreover, the LF–osteopontin complex promoted the proliferation and differentiation of gut cells to a greater extent than the individual proteins. These findings can be of great value in designing infant formula, supplemented with both proteins to enhance their respective bioactivities. 

LF regulation of the innate immune system supports its significance as the host’s first-line defense against pathogens. LF exerts this activity through cellular and molecular mechanisms. At the cellular level, LF receptors have been found on lymphocytes, macrophages, and dendritic cells, being involved in cellular responses such as migration, maturation, and proliferation. At the molecular level, LF interacts with a wide variety of compounds, both soluble and membrane molecules, involved in the inflammatory response. Several studies in animal and human models have shown that LF can prevent and treat gastrointestinal inflammatory diseases by inhibiting the production of pro-inflammatory cytokines, stimulating the synthesis of anti-inflammatory cytokines, and scavenging reactive oxygen species. In addition to modulating innate immunity, LF also exerts a profound modulatory action on the adaptive immune system, connecting both immune responses.

LF is the main component of neutrophil secondary granules and is released by degranulation during inflammation, leading to an increase in the concentration of LF in feces. Consequently, as with calprotectin, LF is being used as a fecal biomarker of intestinal inflammation in some processes, such as ulcerative colitis and Crohn’s disease, as there are some commercial tests based on antibodies for both proteins.

The use of LF to alleviate some intestinal disorders in infants, such as late-onset sepsis and necrotizing enterocolitis in low-birth-weight infants and newborns, has been assayed. However, although some results may indicate a certain improvement in the clinical patients, the evidence is not strong enough to support the routine use of LF to prevent late-onset infection and necrotizing enterocolitis and the associated mortality in preterm infants.

The activity of LF against certain pathogens causing enteric diseases, such as pathotypes of *E. coli*, *C. difficile*, rotavirus, and norovirus, has been shown to be positive in in vitro and in vivo assays. The activity of LF against bacteria can be due not only to the ability of LF to sequester iron, but also to its capacity to block the mechanisms used by the bacteria to adhere to host cells and to invade them, to degrade bacterial toxins, or even to interfere with the expression of virulence factors.

Finally, the efficacy of LF as a selective modulator of the microbiome has to be stressed, as it has been confirmed in several animal models and human studies. Thus, the administration of native and iron-saturated bLF positively modulates the intestinal microbiota of mice with intestinal dysbiosis induced by clindamycin, increasing the levels of Bacteroidaceae, Prevotellaceae, and Rikenellaceae. On the other hand, it has been widely reported that bLF supplementation in the diet of piglets can promote the growth of beneficial microbes, such as *Lactobacillus* and *Bifidobacterium*. Moreover, early life LF intervention in suckling piglets increases the abundance of Bacteroidetes and decreases the levels of Proteobacteria and Fusobacteria. With respect to the role of LF in the modulation of human intestinal microbiota, few studies support it. Talactoferrin (TLF), a drug designation for rhLF, modulates Enterobacteriaceae in the feces of very-low-birth-weight infants, and enteral TLF prophylaxis lowers the burden of *Enterobacter hormaechei*, providing a possible mechanism for a reduction in necrotizing enterocolitis. In this field, more studies are needed to determine the effect of LF supplementation in microbiota, considering the complex variety of microorganisms present and the influence that intestinal microbiota has on the health of humans and animals. 

## Figures and Tables

**Figure 1 pharmaceutics-15-01569-f001:**
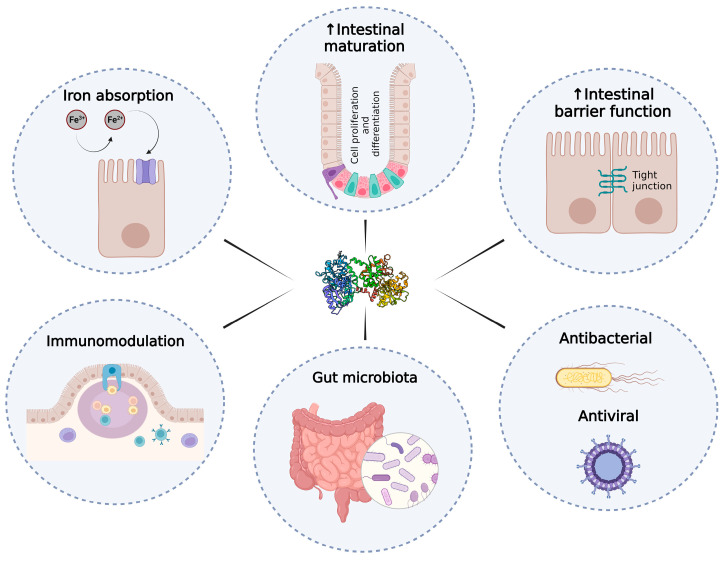
Outline of the potential benefits of LF at the intestinal level. LF can fullfil different roles in the intestine including iron absorption, intestinal cell proliferation and maturation, intestinal barrier enhancement, immunomodulatory activity, antimicrobial activity, and modulation of gut microbiota. Increase (↑).

**Figure 2 pharmaceutics-15-01569-f002:**
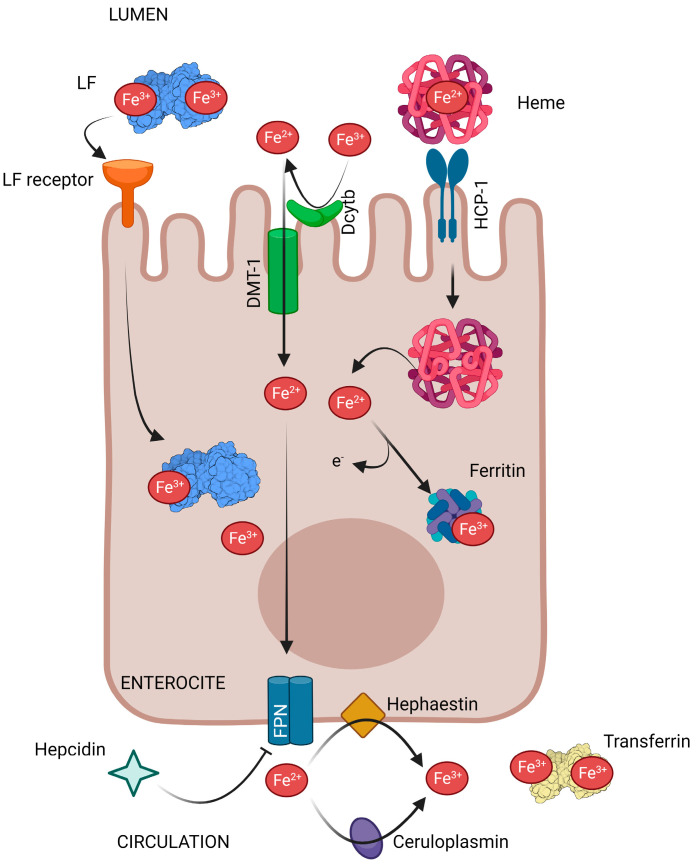
Mechanisms of iron absorption in the enterocyte. Heme Fe^2+^ ions (from meat) enter into enterocyte by the heme carrier protein 1 (HCP-1). Non-heme Fe^3+^ ions need to be reduced to Fe^2+^ ions by Dcytb, prior to being transported into cells by the DMT-1 transporter. Other pathways that may contribute to iron absorption include the LF/LF receptor-mediated iron uptake. Once inside the cell, Fe6^2+^ ions can be stored in the iron-binding ferritin or released into circulation by the ferroportin transporter (FPN). Then, Fe^2+^ ions can be oxidized to Fe^3+^ ions by hephaestin or serum ceruloplasmin, and iron can be transported to tissues by transferrin. The hepatic, hepcidin, as a master regulator of iron, inhibits the activity of FPN and iron release into circulation.

**Figure 3 pharmaceutics-15-01569-f003:**
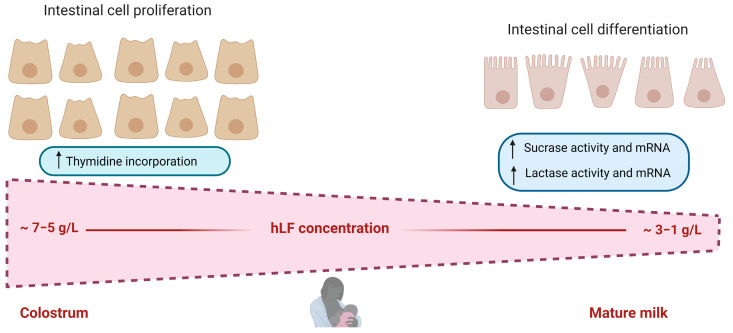
Schematic representation of the effects of human LF (hLF) on small intestinal proliferation and differentiation in early life stages in children. High concentrations of LF, achieved at the first period of lactation in colostrum, enhance intestinal cell proliferation with an increase in thymidine incorporation, whereas low concentrations of LF in mature milk stimulate intestinal differentiation, increasing the sucrase and lactase activities, as well as their mRNA. Increase (↑).

## Data Availability

The data presented in this study are available on request from the corresponding author. The data are not publicly available due to reasons of privacy.

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
