# Peer review of "The Role of Lactoferrin in Intestinal Health"

_pharmaceutics, 2023, doi:10.3390/pharmaceutics15061569_

Round 1
Reviewer 1 Report
the research framework is very interesting and the authors makes effort to describe the influence of the protein lactotransferrin and its the intestinal level in human infants and adults and animal models, on iron intestinal absorption, intestinal growth, maturation and damage repair, regulation of the innate and adaptive immune system, antimicrobial activity and effect on the microbiota, the selected goals were successfully fulfilled. every paragraph is well described and proper conclusion were made. but i have suggestions about references the total number of references is 232 which is in my opinion too much even for review paper especially when one third is older than 2010, hence i suggest authors to summarize references and remove the redundant one.
in my opinion the English style and grammar are fine the paper is easy to understand and read.
Author Response
The number of references has been reduced to 203, eliminating some of the older and redundant ones.

Reviewer 2 Report
Thank you so much for inviting me to review the review paper written by Dr. Conesa et al. Entitled “The role of lactoferrin in intestinal health”. In the study, authors aim to rovide a compilation of all the information related to LF and intestinal health, in infants and adults, since the iron-binding glycoprotein lactoferrin (LF) is one of key components of the mother’s milk. In general, the manuscript is in good written. I have some comments to authors:
(1) In the introduction part, authors mentioned human LF and animal LF, are there obvious difference between human and animal LF, e.g. MOA, structure, functions, etc.?
(2) For receptor of LF, whether Intelectin is the sole receptor of LF?
(3) I suggest that authors could draw a or several schematic diagrams to illustrate how LF involves iron absorption (including MOAs), intestinal growth, maturation and damage repair, the intestinal immune system, some intestinal disorders caused by bacteria, Modulatory effects of lactoferrin on gut microbiota, which will make audience to understand easily.
The writting is fine, minor revision.
Author Response
In the study, authors aim to provide a compilation of all the information related to LF and intestinal health, in infants and adults, since the iron-binding glycoprotein lactoferrin (LF) is one of key components of the mother’s milk. In general, the manuscript is in good written. I have some comments to authors:
(1) In the introduction part, authors mentioned human LF and animal LF, are there obvious difference between human and animal LF, e.g. MOA, structure, functions, etc.?
There are some differences in LF activity among species, but the homology is very high between them and the same type of functions have been identified for all. We have not gone deeper into the differences between species as the review was already very extensive.
(2) For receptor of LF, whether Intelectin is the sole receptor of LF?
Intelectin is not the only receptor to which LF binds as it has been indicated in the section 4.1.
(3) I suggest that authors could draw a or several schematic diagrams to illustrate how LF involves iron absorption (including MOAs), intestinal growth, maturation and damage repair, the intestinal immune system, some intestinal disorders caused by bacteria, Modulatory effects of lactoferrin on gut microbiota, which will make audience to understand easily.
Some of the diagrams suggested have been created to illustrate the potential benefits of LF (Fig. 1), the mechanisms of enterocyte iron absorption (Fig. 2) and the effects of LF on small intestine proliferation and differentiation (Fig. 3).
Comments on the Quality of English Language
The writing is fine, minor revision.
The writing has been improved.

Reviewer 3 Report
The manuscript “The role of lactoferrin in intestinal health” by Celia Conesa et al is a well written, updated ant complete review. The review could benefit from a scheme summarizing LF multidirectional actions towards immune system, pathogens, symbiotic microbiota, inflammation, etc.
minor corrections
in the whole manuscript the Latin names should be written in italics
the authors should decide whether to write “hours” or “h” in the table
Author Response
The manuscript “The role of lactoferrin in intestinal health” by Celia Conesa et al is a well written, updated ant complete review. The review could benefit from a scheme summarizing LF multidirectional actions towards immune system, pathogens, symbiotic microbiota, inflammation, etc.
Minor corrections:
- In the whole manuscript the Latin names should be written in italics
The Latin names have been written in italics.
- The authors should decide whether to write “hours” or “h” in the table
The term hours has been abbreviated to “h” in the table.

Reviewer 4 Report
The primary objective of this manuscript is the impact of lactoferrin on gut health. This literature review provides an overview of the effects of lactoferrin on gut cell development and integrity, as well as the absorption and regulation of iron by lactoferrin. Additionally, the review discusses the immunomodulatory effects of lactoferrin and its anti-inflammatory properties. Furthermore, the review briefly describes the effects of lactoferrin on gut microbiota. The review is well-written and contains abundant details. Although the author's original intention was to describe and discuss all the protective or regulatory effects of lactoferrin on the gut, the description of lactoferrin's regulation of gut microbiota is somewhat insufficient. In fact, I recommend that the author exclude content related to lactoferrin's regulation of gut microbiota, as recent publications have already extensively described this topic, including the regulation of gut microbiota by lactoferrin and the piglet model (Int. J. Mol. Sci. 2019, 20, 4707; doi:10.3390/ijms20194707 The Impact of Lactoferrin on the Growth of Intestinal Inhabitant Bacteria). I suggest that the author cite this review and remove duplicated or similar content. (Int. J. Mol. Sci. 2019, 20, 4707; doi:10.3390/ijms20194707 The Impact of Lactoferrin on the Growth of Intestinal Inhabitant Bacteria)
Minor part:
Line 43- 45 : the authors indicate that: The aim of this review has been to compile the effects that LF can exert at the intestinal level in human infants and “adults” and animal models, including: iron intestinal absorption, intestinal growth, maturation and damage repair, regulation of the innate.
What’s the lactoferrin roles in adults? In general, it is less likely that adults can receive abundant amounts of lactoferrin because lactoferrin cannot not be obtained from the “natural diets” (only baby can receive enough lactoferrin from the colostrum). Although LF administration (or LF supplement) can help to deal with pregnant women suffering from ID and IDA but this is not the case in normal physiology. Please clarify or revise.
Author Response
The primary objective of this manuscript is the impact of lactoferrin on gut health. This literature review provides an overview of the effects of lactoferrin on gut cell development and integrity, as well as the absorption and regulation of iron by lactoferrin. Additionally, the review discusses the immunomodulatory effects of lactoferrin and its anti-inflammatory properties. Furthermore, the review briefly describes the effects of lactoferrin on gut microbiota. The review is well-written and contains abundant details. Although the author's original intention was to describe and discuss all the protective or regulatory effects of lactoferrin on the gut, the description of lactoferrin's regulation of gut microbiota is somewhat insufficient. In fact, I recommend that the author exclude content related to lactoferrin's regulation of gut microbiota, as recent publications have already extensively described this topic, including the regulation of gut microbiota by lactoferrin and the piglet model (Int. J. Mol. Sci. 2019, 20, 4707; doi:10.3390/ijms20194707 The Impact of Lactoferrin on the Growth of Intestinal Inhabitant Bacteria). I suggest that the author cite this review and remove duplicated or similar content. (Int. J. Mol. Sci. 2019, 20, 4707; doi:10.3390/ijms20194707 The Impact of Lactoferrin on the Growth of Intestinal Inhabitant Bacteria)
We agree that the effect of lactoferrin on certain intestinal microorganisms has been extensively studied in vitro; however, the role of lactoferrin on the regulation and interactions of the entire intestinal microbiota in vivo is still poorly known. For this reason, according to the reviewer, we have included the 2019 review and we have focused on highlighting those studies that show the regulation of the microbiota by lactoferrin in vivo, especially those published after 2019 that are not included in that review.
Minor part:
Line 43- 45: the authors indicate that: The aim of this review has been to compile the effects that LF can exert at the intestinal level in human infants and “adults” and animal models, including: iron intestinal absorption, intestinal growth, maturation and damage repair, regulation of the innate.
What’s the lactoferrin roles in adults? In general, it is less likely that adults can receive abundant amounts of lactoferrin because lactoferrin cannot not be obtained from the “natural diets” (only baby can receive enough lactoferrin from the colostrum). Although LF administration (or LF supplement) can help to deal with pregnant women suffering from ID and IDA but this is not the case in normal physiology. Please clarify or revise.
We agree that the only natural food with a relevant content of lactoferrin is human milk (not only colostrum, as the concentration in mature milk is also relevant). Thus, many of the investigations performed on lactoferrin are aimed to know which would be the benefits of this protein for adults, when added as a supplement to certain foods or administered with a therapeutic purpose. We have tried to show this approach with the different studies included in the review.

Round 2
Reviewer 4 Report
I am satisfied with the author's response.